# A Simple and Safe Protocol for Intra-Testicular Gene Delivery in Neonatal Mice Using a Convenient Isoflurane-Based Anesthesia System

**DOI:** 10.3390/biotech14040081

**Published:** 2025-10-22

**Authors:** Kazunori Morohoshi, Miho Ohba, Masahiro Sato, Shingo Nakamura

**Affiliations:** 1Division of Biomedical Engineering, National Defense Medical College Research Institute, 3-2 Namiki, Tokorozawa, Saitama 359-8513, Japan; 2Department of Genome Medicine, National Center for Child Health and Development, Tokyo 157-8535, Japan; sato-masa@ncchd.go.jp

**Keywords:** newborn, genome-engineering, inhalation anesthesia system, isoflurane, testis, intra-testicular gene delivery, anesthetic device

## Abstract

Newborn mice (up to 6 d after birth) are suitable for genetic manipulations, such as facial vein-mediated injection, owing to their hairless and thin skin. Their small body volumes also facilitate the rapid dissemination of injected solutions, supporting gene engineering-related experiments. However, anesthesia in newborns is challenging because of the potential risks associated with anesthetic agents. Isoflurane inhalation anesthesia is an option, although its effects on brain development remain under investigation. In this study, we established a reproducible protocol for delivering nucleic acids to juvenile mouse testes using a simple isoflurane-based anesthetic system prepared from common laboratory equipment. Using this system, nucleic acids were successfully delivered to juvenile mouse testes via intra-testicular injection, followed by in vivo electroporation. The present isoflurane-based method achieved >90% postoperative survival with normal maternal nursing observations. Gene delivery resulted in limited transfection of seminiferous tubules but efficient interstitial Leydig cell transfection. Thus, gene engineering in somatic and germ cells in neonatal mice will be facilitated using the anesthetic protocol established in this study.

## 1. Introduction

Genetically engineered (GE) mice are useful for studying the functions of specific genes in vivo as well as for developing human disease models and drug discovery. The recent advent of genome editing techniques, such as CRISPR/Cas9, has made it possible to generate these mice in a relatively short time (within a month) [1]. For example, pronuclear microinjection of genome editing reagents at the zygotic stage or in vitro electroporation (EP) of zygotes in the presence of these reagents are now frequently employed to generate GE mice [2,3,4]. Direct introduction of genome editing reagents into the lumen of the oviduct of a pregnant female (at the late zygotic stage) and subsequent in vivo EP of the entire oviduct, called *i*-GONAD, is also considered an efficient method for generating GE mice and rats [5,6]. Notably, all these techniques focus on gene delivery to early embryos, such as fertilized eggs (zygotes). On the other hand, an attempt to deliver genes into male germ cells, including mature sperm or spermatogenic immature cells, also known as spermatogonia, is still challenging. Furthermore, genetic modification of spermatogonial stem cells in vitro followed by their transplantation into the recipient testis is a robust and widely practiced technique for generating germline-edited animals [7]. Although a successful method for generating GE murine fetuses, known as sperm-mediated gene transfer (SMGT), has recently been reported, this approach remains complex and is still challenging. The technique involves performing in vitro fertilization by mixing and oocyte in the presence of genome editing reagents and methyl β-cyclodextrin, a reagent that facilitates DNA association with the sperm surface before embryo transfer [8].

Neonatal mice up to 6 d of age lack body hair and the skin is thin, which facilitates the visualization of blood vessels (i.e., facial vein) and internal organs under a stereomicroscope. These features allow drug administration through facial veins or direct surgical approaches to target internal organs. In addition, their small compact body volume facilitates expeditious dissemination of the injected solution throughout the body, thereby achieving the desired therapeutic effects. Thus, neonatal mice are considered an ideal target for administering genome editing reagents using various approaches, such as gene delivery via the facial vein and direct injection into target tissues or organs under appropriate anesthesia. This approach is called “in vivo somatic cell gene modification” and can offer an alternative to generating germline-edited animals via embryo manipulation [9].

The surgical manipulation of neonatal mice under a stereomicroscope presents several technical challenges due to their inherent vulnerabilities. One of the key considerations in this context is the administration of controlled anesthesia. Hypothermia-based anesthesia is commonly employed to induce unconsciousness in neonatal mice by safely lowering their body temperature, typically achieved through placement on ice [10]. This technique is often employed in short surgical procedures and does not require specialized facilities. However, it is difficult to control the depth of anesthesia, which often causes unexpected awakenings. Furthermore, there is a risk of cardiac arrest due to excessive hypothermia [11]. In addition, sudden hypothermia causes physiological stress in mice, leading to delayed recovery after surgery, which often results in maternal abandonment or cannibalism and significantly decreases survival rates. Therefore, hypothermic anesthesia may not be suitable for invasive and precise surgical treatments. Isoflurane-based anesthesia is also widely used to anesthetize neonatal mice [12]. However, the potential side effects, especially in the developing brain, must be carefully considered. Thus, prolonged exposure to isoflurane may cause hypoglycemia and neurodegeneration [13,14,15]. Furthermore, specialized equipment is required for induction, maintenance, and recovery. This equipment includes a calibrated vaporizer, nose cone or mask, and a scavenging system to minimize personnel exposure. When anesthetizing neonatal mice, it is important to adjust the mouse’s nose to the nozzle from which isoflurane is released. Ho et al. [12] recently developed special equipment comprising laser-formed plastic structures using an inhalation anesthesia approach to anesthetize neonatal mice. The system utilizes a specialized anesthesia mask, produced through 3D printing technology, to securely stabilize the mouse’s head during procedures. This approach facilitates precise injection controlled by a micromanipulator. However, it is important to note that this method necessitates access to specific equipment and a high quality 3D printing facility, which can be costly.

Therefore, the primary objective of this study was to establish a simple and reproducible protocol for performing genetic manipulation of testes from highly vulnerable neonatal mice (days 3–5). To achieve this, a critical step was to develop a convenient and safe anesthetic system that overcomes the limitations of both hypothermia and existing specialized inhalation devices. Using this system, we successfully delivered nucleic acids (plasmid-based) into germ cells and other somatic cells in neonatal testes through intra-testicular injection and subsequent in vivo EP. It took <30 min to complete gene delivery for both testes.

## 2. Materials and Methods

### 2.1. Animals

B6C3F1 and ICR mice (8–10 weeks of age; Japan SLC, Inc., Shizuoka, Japan) were used in all experiments. The mice were maintained under controlled temperature conditions with a 12 h light/dark cycle. The females were naturally mated with males, and the day of birth was designated as day 0 of pregnancy. Food and water were provided ad libitum. Neonatal mice on days 3–5 were subjected to surgery, as described below. All animal experiments were approved by the Care and Use of Laboratory Animals at the National Defense Medical College (permit no. 23053) and conducted per the guidelines of the National Defense Medical College Committee on Recombinant DNA Security (permit no. 2024-11).

### 2.2. Sex Determination of Neonatal Mice

To determine the sex of neonatal mice on days 3–5, the detection of pigmented spots present between the genitalia and anus is useful in pigmented mice such as B6C3F1. These pigmented spots were visible in male (but not female) neonates (arrows in Figure 1a vs. Figure 1b). However, these pigmented spots were difficult to detect in albino neonates such as ICR (Figure 1c). To determine the sex of neonatal ICR mice, we checked for the presence or absence of nipples on their abdominal skin after inspection under a dissecting microscope.

### 2.3. Plasmid DNA

The plasmid used in this study was an expression plasmid (pAQI; gifted by Prof. Masato Ohtsuka, Tokai University, School of Medicine, Isehara, Japan), in which *tdTomato* cDNA expression was controlled by the ubiquitous and strong promoter CAG [16]. This plasmid was amplified after transformation into DH5α competent cells (BioDynamics Laboratory Inc., Tokyo, Japan) and purified using a Plasmid Maxi kit (Qiagen, Venlo, the Netherlands). The purified DNA was dissolved in phosphate-buffered saline (PBS) + 0.02% (*v*/*v*) Fast Green FCF (Nacalai Tesque, Kyoto, Japan) to a final concentration of 0.25 μg/μL.

### 2.4. Convenient Inhalation Anesthesia System for Neonatal Mice

An anesthesia chamber with a 2.0 L/min flow of 3% isoflurane in O_2_ was used for rapid anesthetic induction before animal surgery (Figure 2a). It usually takes only 2–5 min for spontaneous movement to disappear. To apply anesthesia devices commonly used in the laboratory to neonatal mice, the tip of a rubber glove (or rubber finger, purchased from Ansell Ltd., Richmond, Australia) was cut off to create an opening (Figure 2b,c). The neonatal mouse’s nose was inserted through the opening, from which a 3% isoflurane is constantly provided (Figure 2d,e). Alternatively, it is also useful for anesthetizing a neonatal mouse using a 15 mL centrifuge tube containing cotton wool soaked with isoflurane anesthetic and capped with the cut tip of a rubber finger (Figure 2f,g). We typically started with an initial volume of approximately 100 μL on the cotton wool. The anesthetic depth was then assessed by monitoring the pup’s response to stimuli (including a firm toe pinch). Small increments of isoflurane were added as needed until the pedal withdrawal reflex was completely absent.

### 2.5. Intratesticular Injection-Based Gene Transfer (IIGT)

IIGT was performed, based on our previous paper [17] shown for the adult murine testis. As shown in Figure 3a, the intra-testicular injection of DNA solution was first directed towards the exposed testis. Subsequently, in vivo EP was performed on all testes. In Figure 3b–l, more detailed procedures are shown. Briefly, a small incision was made using microscissors in the lower portion of the abdominal skin of a neonatal male mouse anesthetized using the aforementioned method (Figure 3b). The muscle layer beneath the incision was excised (Figure 3c). One testis was removed through an incision on the abdominal wall (Figure 3d). Under a dissecting microscope, a solution (2.5 μL) containing plasmid DNA and Fast Green FCF was directly injected into the testis using a glass capillary attached to a mouthpiece (Figure 3e–h). Glass capillaries (Drummond Scientific Company, Broomall, PA, USA) were created by pulling through a pipette puller (Narishige, Tokyo, Japan) using a single-step heating protocol. The tips were opened with forceps to make the inner diameter 40–60 μm. Immediately after injection, the entire testis was covered with a small piece of wet paper (KimWipe; Nippon Paper Crecia, Tokyo, Japan) after dipping in PBS and then held with tweezers-type electrodes (Figure 3i). In vivo EP was performed using a square-pulse generator (CUY Edit II; BEX Co., Tokyo, Japan) with the following EP parameters (ten square-wave pulses with a pulse duration of 50 ms and an electric field intensity of 50 V). After EP, the testis returned to its original position (Figure 3j). The other testes were also exposed to the abdominal skin and subjected to DNA delivery and subsequent in vivo EP as described above. After completing EP for both testes, the abdominal wound was closed using sutures (Figure 3k,l). Toe clipping was performed to distinguish between mice that underwent surgery and untreated intact mice.

### 2.6. Recovery

Part of the nest was moved from the home cage to a small box, which was placed on a thermostatically controlled heating pad before starting the procedure (Figure 2h). The total duration of anesthesia was 30 min. After the surgical procedure, the treated mice were placed in a warm box for at least 5 min for recovery (Figure 2i). The recovered mice were then moved to their home cage with the nest after confirming that they exhibited spontaneous movement (Figure 2j). In general, maternal mice exhibit a behavior known as infanticide, which involves consuming their offspring, particularly instances where there are sick or deformed pup, or when pups have succumbed naturally. The effectiveness of our isoflurane-based anesthetic system was evaluated by monitoring the survival rate of treated mice one day following surgical procedures.

### 2.7. Fluorescence Microscopy

Two days after in vivo gene delivery, all 42 surviving neonatal mice were sacrificed by cervical dislocation for fluorescence analysis. Electroporated testes were excised and subjected to mechanical removal of the tunica albuginea. The seminiferous tubules (STs) were separated from the surrounding tissue by mechanical disruption using microscissors under a dissecting microscope to liberate their highly organized structures on a plastic dish. After covering the released STs with a coverslip, tdTomato-derived fluorescence was examined using an all-in-one fluorescence imaging microscope (BZ-X800; Keyence, Tokyo, Japan).

### 2.8. Statistical Analysis

The post-operative survival rate is presented as a percentage. The 95% confidence interval (CI) for the overall survival rate was calculated for the pooled data (42 survivors out of 45 pups) using the Clopper-Pearson (exact) method based on the binomial distribution.

## 3. Results

### 3.1. Establishment of a Stable Inhalation Anesthesia System for Neonatal Mice and a High Survival Rate After Surgery

We administered intra-testicular injections in neonatal mice on days 3–5 using a convenient inhalation anesthesia system that we developed originally. DNA injection into both testes of the neonate was completed within 30 min. No awakening occurred during surgery, and postoperative awakening was rapid. When the treated mice were returned to their mothers, they were accepted normally, and no abandonment or cannibalism, which may have been caused by anesthesia, was discernible. Of the 45 pups that were subjected to IIGT, 42 (93%; 95% confidence interval: 83.0–98.5%) successfully survived (Table 1).

### 3.2. Gene Transfer in the Neonatal Testis

When fluorescence in STs together with interstitial cells within the testis was assessed 2 d after intra-testicular gene delivery, tdTomato-derived fluorescence was detected in a large number of cells over a wide area in the interstitial region of the testis (left testis in Figure 4a,b). No fluorescence was observed in the uninjected control testes (right testis in Figure 4a,b). When STs were partially separated from the surrounding tissue by mechanical disruption to liberate their highly organized structures, the fluorescent cells were identified as Leydig cells, based on their morphology and localization (Figure 4c,d). In contrast, the vast majority of the STs were negative for tdTomato-derived fluorescence. Only a few cells, probably SSCs localized inside the STs, exhibited fluorescence (arrows in Figure 4e–g). It took <30 min to complete gene delivery for both testes.

## 4. Discussion

In this study, we established a simple and safe protocol for in vivo gene delivery to the testes of neonatal mice, achieving a high postoperative survival rate of over 90%. The major achievement of this method is the successful application of an invasive surgical procedure to a highly vulnerable age group (on days 3–5), providing a new platform for genetic manipulation in juvenile mice.

The key to this success was our convenient isoflurane-based anesthetic system, which is simpler and safer than traditional hypothermic anesthesia, often associated with risks of cardiac arrest and maternal cannibalism. While the usefulness of an inhalation anesthesia approach in neonatal mice has been reported previously [12,18], our study provides crucial advancements in practical application and reproducibility. For instance, the system reported by Ho et al. [12] allows for precise injections but requires specialized, 3D-printed equipment. The earlier report by Gotoh et al. [18] on mice aged 6–10 d did not adequately describe the detailed procedure during the anesthetic period and subsequent recovery phase, which could hinder its effective application. In contrast, our system, based on readily available laboratory consumables, provides a detailed, step-by-step protocol that ensures high reproducibility. This makes our method highly accessible and practical for a wide range of laboratories aiming to perform delicate surgeries on neonatal mice.

Our method for delivering genes to neonatal testes builds upon previous work, such as that by Ju et al. [19], who first demonstrated transgenesis via intra-testicular injection in neonates. Their study focused on the functional outcome of germline transmission, reporting a transgenesis rate of up to 14.3% (average 12.5% in their optimal low-dose group) in the F1 offspring generated from the treated males. However, their seminal report, while demonstrating the possibility of germline transmission, lacked detailed information on the post-operative survival rate of the manipulated neonatal pups themselves. This is a critical gap that our study addresses, as a high procedural survival rate is a prerequisite for any subsequent functional studies or breeding programs. We demonstrate that this challenging surgery, applied to an even younger and more vulnerable age group (days 3–5 vs. day 7 in Ju et al.’s study [19]), can be achieved with a remarkably high and statistically robust survival rate (>90%; 95% CI: 83.0–98.5%). We credit this achievement to our carefully developed and consistent anesthetic protocol, which effectively mitigates the risk associated with hypothermia. This approach creates a reliable and secure foundation for conducting genetic interventions during a crucial early post-natal period.

Notably, previous studies have indicated that administering inhalational anesthetic agents such as isoflurane and sevoflurane at the juvenile stage may promote apoptosis of the developing cranial nervous system and impair learning and memory [13,14,15]. However, neurotoxicity can be reduced by preconditioning with hydrogen gas or treatment with specific agents before anesthesia [20,21]. Based on this background, a careful experimental design (i.e., shortening the duration of anesthesia as much as possible) is required, especially when applying this approach to study brain function and behavior.

In mice, SSCs exist within the STs and have a unique ability to self-renew and differentiate into sperm cells [22]. Gene delivery to these cells can be a useful tool for generating GE animals. For example, this can be achieved by the direct injection of DNA into the testis and subsequent in vivo EP [23,24,25] or in vivo transfection of SSCs after retrograde gene delivery via a rete testis [26,27,28]. Unfortunately, little is known about gene delivery to juvenile testes at the neonatal stage (3–7 d after birth). To our knowledge, Ju et al. [19] were the first to demonstrate that intra-testicular injection of plasmid DNA encapsulated in liposomes resulted in the generation of transgenic offspring when these gene-delivered individuals were subjected to natural mating with estrous females. Although they did not demonstrate the mechanism by which exogenous DNA introduced inside the testis can be transmitted to the offspring, SSCs or immature sperm cells likely take up DNA. Ju et al. [19] suggested that the intra-injection of plasmid DNA and subsequent in vivo EP can result in the efficient transfection of SSCs or related cells. Our experiments demonstrated that transfection of interstitial cells, as exemplified by Leydig cells, in the testis was prominent, but only a few successfully transfected cells existed within STs. In this context, retrograde gene delivery via the rete testis may be a better route for accessing SSCs, although application of this technique to juvenile testes is very difficult.

While our protocol proved effective, we recognize several limitations that warrant discussion. It should be noted that our assessment of gene transfer was qualitative, confirming the presence and location of transfected cells. A quantitative analysis of transfection efficiency, for instance, by flow cytometry or quantitative polymerase chain reaction, was beyond the scope of this protocol-development study but represents an important next step for future work. Regarding scalability, the procedure takes approximately 30 min per animal, which may limit the throughput for large-scale screening experiments. In terms of strain variability, our study successfully utilized B6C3F1 mice, but the safety and efficacy of this protocol should be validated in other mouse strains. Finally, the surgical procedure itself is technically demanding and operator-dependent. A learning curve should be anticipated for researchers new to this delicate technique.

## 5. Conclusions

This study yields two important results. The first is the establishment of a safe and reproducible protocol for performing intra-testicular gene delivery in highly sensitive day 3–5 neonatal mice with a high survival rate. This was enabled by our development of a convenient inhalation anesthesia device that can be fabricated easily using readily available laboratory consumables. The second is the demonstration that this method allows for the delivery of exogenous DNA to interstitial cells, such as Leydig cells, in juvenile murine testes. This protocol can be used in a wide range of experiments involving genetic manipulation in neonatal mice, offering a valuable tool for developmental and reproductive biology.

## Figures and Tables

**Figure 1 biotech-14-00081-f001:**
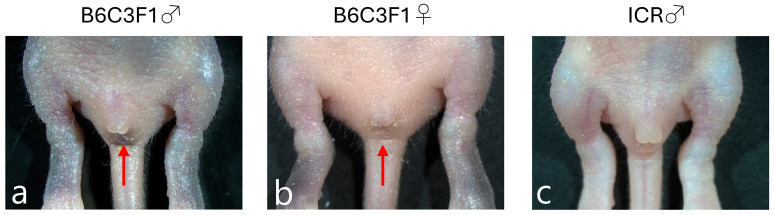
Sex determination of neonatal mice. The neonatal male B6C3F1 mice can be identified by the presence of pigmented spots (shown by arrow) located between the anus and the genitalia (**a**). The neonatal female B6C3F1 mice do not show such a phenomenon ((**b**), shown by arrow). In albino male mice, such as ICR, these pigmented spots are difficult to see (**c**). The presence/absence of nipples in the abdominal albino skin is an important point in determining the sex.

**Figure 2 biotech-14-00081-f002:**
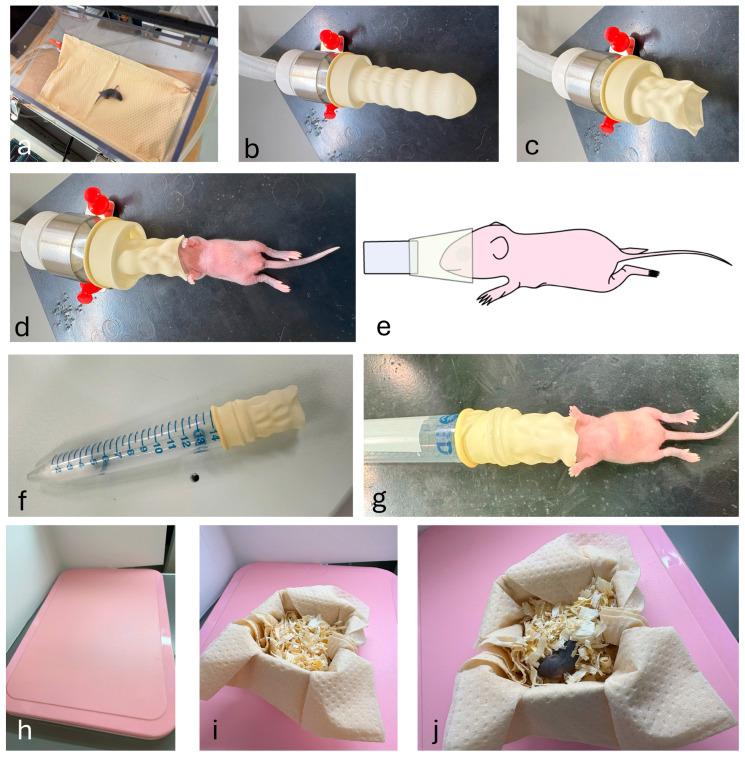
Procedure for inducing and maintaining inhalation anesthesia in neonatal mice. (**a**–**d**) Convenient isoflurane-based anesthetic device. Anesthesia was first induced by placing the mouse in an anesthesia chamber. Then, the anesthetized mouse is placed onto an experimental platform (made of rubber) by inserting its snout into the opening of a rubber glove finger. The opening is created by cutting off the tip of a rubber glove finger. (**e**) Schematic presentation of the induction of anesthesia in neonatal mice. (**f**,**g**) An alternative isoflurane-based anesthetic device. In this case, a 15 mL conical tube (containing cotton wool soaked with anesthetic at its bottom) can be used. Care must be taken regarding the volume of anesthetic applied to the cotton wool. (**h**–**j**) Treatment after surgery. To recover the anesthetized mouse, it is placed in a box with a part of the nest from the home cage. This provides familiar scents to minimize stress and potential cannibalism. The entire box was placed on a controlled heating plate.

**Figure 3 biotech-14-00081-f003:**
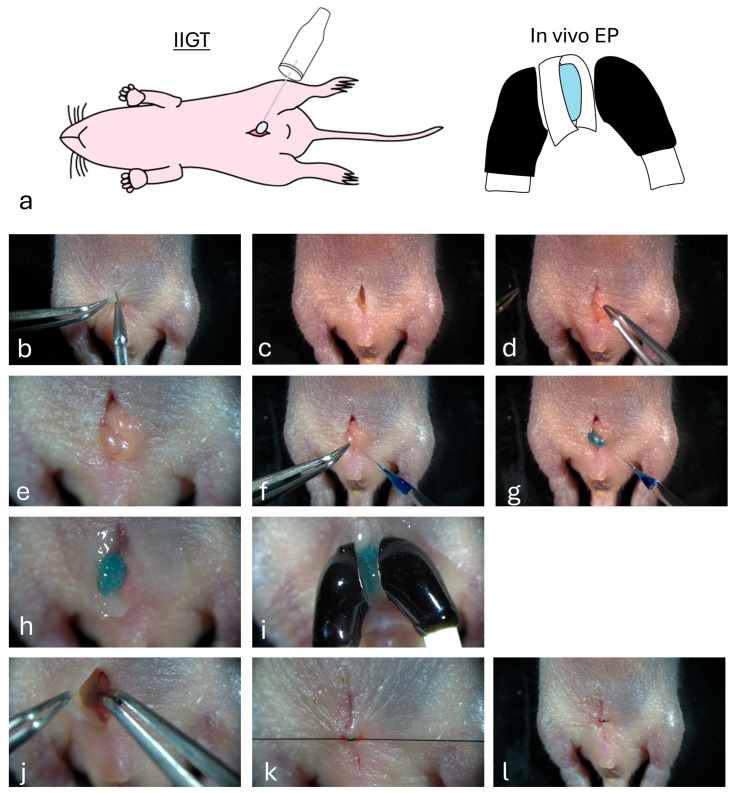
Surgical procedure for gene delivery to a juvenile testis. (**a**) Schematic presentation of intra-testicular injection-based gene transfer (IIGT) [intra-testicular injection and subsequent in vivo electroporation (EP)]. (**b**–**d**) An abdominal incision was made using microscissors to expose the mouse testis. (**e**–**g**) Intra-testicular injection of DNA-containing solution using a glass capillary. (**h**,**i**) In vivo EP. After injection, the entire testis is covered with a small piece of KimWipes soaked in buffer, and then subjected to EP by grasping it with tweezer-type electrodes. (**j**–**l**) Skin closure. The abdominal incision is closed using sutures.

**Figure 4 biotech-14-00081-f004:**
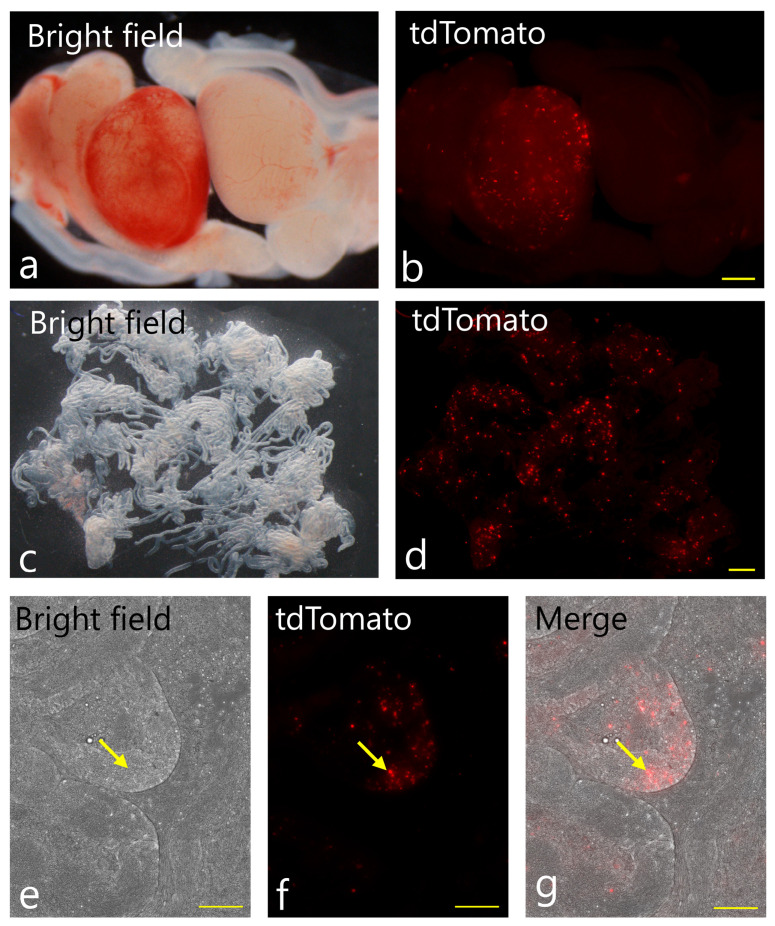
Gene delivery and gene expression in neonatal murine testis. (**a**,**b**) Representative images for in vivo EP-treated (**left**) and non-treated control (**right**) testes of the same mouse. In the left panel, a photograph was taken under the light. In the right panel, a photograph was taken under epi-fluorescence illumination. Note the presence of bright fluorescent spots in the inner area of a testis into which gene delivery is performed. In contrast, no fluorescence is discernible in the control testis. Bar: 500 µm. (**c**,**d**) Seminiferous tubules (STs) of the in vivo EP-treated testis were dissociated mechanically. In the left panel, a photograph was taken under the light. In the right panel, a photograph was taken under epi-fluorescence illumination. The fluorescent signal is prominent in the cell clusters (probably Leydig cells) localized in the interstitial space of STs. Bar: 500 µm. (**e**–**g**) Fluorescence image of the in vivo EP-treated STs. Some STs exhibited tdTomato-derived fluorescence in the internal area of STs (shown by arrows). Photographs were taken under light (**e**) or epi-fluorescence illumination (**f**). (**g**) shows merge from (**e**–**f**). Bar: 100 µm.

**Table 1 biotech-14-00081-t001:** Summary of in vivo gene delivery (intratesticular injection-based gene transfer, IIGT) towards juvenile testis.

No. of Experiment	Total No. of Newborns Subjected to IIGT	No. Pups Died or Killed By a Mother	No. Pups Survived ^1^
1	9	1 ^2^	8
2	13	1 ^2^	12
3	10	0	10
4	7	0	7
5	6	1	5
total	45	3 ^1^	42
success rate			93% (95% CI: 83.0–98.5%)

^1^ Survival of pups was determined by visual inspection for their alive one day after IIGT. ^2^ Heavy bleeding during surgery was observed.

## Data Availability

The original contributions presented in this study are included in the article. Further inquiries can be directed to the corresponding authors.

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
