# Peer review of "A Simple and Safe Protocol for Intra-Testicular Gene Delivery in Neonatal Mice Using a Convenient Isoflurane-Based Anesthesia System"

_biotech, 2025, doi:10.3390/biotech14040081_

Round 1
Reviewer 1 Report
Comments and Suggestions for Authors
The manuscript by Morohoshi et al., describes a body of work in which the research team developed a novel anesthetic methodology to be used on young, neonatal mice to allow genetic manipulation of resident cells in the testis. The method is easy, very practical and will be of significant utility in manipulating cells in the juvenile testis. Well done.
Specific comments:
Introduction:
Line 39- substitute “of” for “towards”.
Line 40- remove “animals such as”
Line 42- remove “towards” and include “deliver genes into”
Line 42- remove “such” and include “including”
Line 43- It is worth noting that the methodology to deliver genes into male germ line stem cells in vitro and the transfer of these cells into the rete testis is a very robust technique practiced worldwide. It should be noted.
Line 48- insert lack “body” hair, remove “throughout the skin. Remove skin “itself”
Line 57- Very awkward sentence, particularly the phrase “can be replaced by….”
Comments on the Quality of English LanguageI made some corrections to the text. Small changes but meaningful.
Author Response
We sincerely thank Reviewer for their very positive and encouraging review, and for recognizing the practical utility of our method. We appreciate the specific comments on the text, which have helped us to improve its clarity.
Comment: The English could be improved to more clearly express the research.
Response: We thank for this suggestion. To improve the language quality, the revised manuscript has been thoroughly edited by a professional English editing service, Editage. We have attached the editing certificate for your reference and are confident that the clarity of the manuscript has now been significantly improved.
Comment (Line 39, 40, 42, 48): Suggestions for word substitutions and removals.
Response: Thank you for these excellent suggestions. We have revised the sentences indicated by the reviewer (Please see lines 41-44, 53 in the revised text).
Comment (Line 43): It is worth nothing that the methodology to deliver genes into male germ line stem cells in vitro and the transfer of these cells into the rete testis is a very robust technique practiced worldwide. It should be noted.
Response: This is a very important point. We have now added the sentence to the Introduction section (Please see lines 45-52 in the revised text).
Comment (Line 57): “Very awkward sentence, particularly the phrase “can be placed by….”
Response: Based on your suggestion, this portion was rewritten (Please see lines 61-62 in the revised text).

Reviewer 2 Report
Comments and Suggestions for Authors
This manuscript, titled "Development of a convenient isoflurane-based anesthetic device useful for performing genetic manipulation in juvenile murine testes," details the creation and application of a simple isoflurane anesthesia system tailored for use with neonatal mice during genetic manipulations of the testes. The work addresses relevant technical challenges and describes a potentially valuable tool for the field. The text is generally well-organized and clear, and the experimental procedures are presented in detail.
Major Concern: Title vs. Content
A major concern is that, despite the manuscript's title emphasizing the development of an anesthetic device/method, almost all the experimental results are focused on the efficiency and procedures of genetic manipulation in the testes. There is limited direct assessment or quantitative data provided regarding the efficacy, safety, or broader physiological effects of the isoflurane-based anesthetic method itself in neonatal mice. Beyond reporting a high postoperative survival rate and general acceptance by the mother, no specific behavioral, physiological, or neurological analyses of the anesthetized pups are presented. As currently presented, the manuscript may give readers the impression that an in-depth evaluation of this anesthesia method is included, which is not the case.
Recommendation:
Either the title should be revised to more accurately reflect the primary focus on genetic manipulation in the testes using a new anesthesia protocol, or the study should be expanded to include more comprehensive data and analysis on the anesthetic method itself—such as including direct comparisons of the new and existing anesthesia methods with respect to induction and recovery times, stress markers, complication rates, or other relevant endpoints and long-term effects on the pups if available.
Please expand your discussion of limitations with regard to scalability, strain variability, and operator dependency.
Minor Comments:
1. Some typographical and grammatical errors are present and could be improved by further language editing.
2. Where possible, include numerical data with statistical confidence intervals.
3. Ensure figure citations are consistent and in order.
Author Response
We thank Reviewer 2 for his/her thorough review and for recognizing our work as a “potentially valuable tool.” We appreciate the specific comments on the text, which have helped us to improve its clarity.
(Major Concern: Title vs. Content)
Comment: “A major concern is that… the manuscript’s title emphasizing the development of an anesthetic device/method, almost all the experimental results are focused on… genetic manipulation… Either the title should be revised… or the study should be expanded…”
Response: As suggested, we found that the original title created a mismatch with the experimental results. As per the reviewer’s excellent recommendation, we revised the title and adjusted the focus of the manuscript to more accurately reflect our primary contribution (Please see lines 2-4, 18-21, 27-28, 88-92, 252-268, 320-328 in the revised text).
Comment: Please expand your discussion of limitation with regard to scalability, and operator dependency.
Response: We thank the reviewer for this valuable suggestion. We have added a new paragraph to the Discussion section to address the limitations of scalability, and operator dependency (Please see lines 308-318 in the revised text).
(Minor comments)
Comment: Some typographical and grammatical errors are present…
Response: The revised manuscript has been thoroughly proofread and professionally edited for English language and clarity.
Comment: Where possible, include numerical data with statistical confidence intervals.
Response: This is an excellent point that aligns with a comment from another reviewer. As requested, we have calculated the 95% confidence interval for our primary outcome, the post-operative survival rate, and have added this information to Table 1 and the corresponding text in the Results section. Furthermore, to ensure transparency and reproducibility, we have added a new subsection titled “statistical analysis” to the Materials and Methods section, which explicitly describes the method used for this calculation (the Clopper-Pearson exact method). (Please see lines 207-211, 221 and Table 1 in the revised text).
Comment: Ensure figure citation are consistent and in order
Response: We have carefully reviewed all figure citations to ensure that they are sequential and consistent.
Reviewer 3 Report
Comments and Suggestions for Authors
This study mainly aimed to develop an inexpensive inhalation anesthetic device for (neonatal) mice and test its feasibility (and methodology) to transfer nucleic acids into germ cells and other somatic cells in neonatal testes under an isoflurane anesthesia protocol. This study is well designed, and the subject presents novelty to be published. Overall, it is well written and supported by relevant figures, one table, and adequate references. Nonetheless, some clarifications and improvements can add value. Some part of M&M was only reported in the results section. It will be adequate to make a sequential mention in M&M. Table 1 can be improved by reporting the 95% confidence interval. It will be possible to quantify and test the differences in the fluorescence test between the treated and control groups? I suggest being more precise in the conclusions. Also, the authors claim the “development of a method to deliver exogenous DNA to interstitial cells” as a conclusion. Please can you discuss (in the discussion section) what is new in this methodology regarding other published experiments?
Specific comments:
L78-79: I suggest reporting the advantages and limitations of this equipment/methodology (L247-249). This information will support the justification for the objectives of your study.
L182: “3-1. Sex determination of neonatal mice”. Can this fit better in M&M?
L206: You report five replicates (experiments) of the study. So, you can calculate the 95% confidence interval. This interval is relevant to define the lower and upper intervals of the success rate.
L217: 2? It is not reported in Table 1.
L210-214: This two-day assessment was made in all 42 pups?
L217: Can you quantify (in percentage) instead of using “almost”?
L237: “(93 %)”.
L274-276: What was the success of these previous experiments?
Author Response
We thank Reviewer 3 for their positive assessment that our study is “well designed” and presents “novelty to be published.” We have addressed all the valuable suggestions for clarification and improvement.
Comment: Some part of M&M was only reported in the result section
Response: As suggested, we moved the subsection on sex determination shown in the previous Results section to the Materials and Methods section. (Please see lines 108-115 in the revised text).
Comment: Table 1 can be improved by reporting 95% confidence interval
Response: This is an excellent suggestion. We have now calculated the 95% confidence interval for the pooled survival rate using the Clopper-Pearson (exact) method and have added this information to Table 1 and the main text. To provide full details on how this was calculated, we have also introduced a new “Statistical analysis” subsection in the Materials and Methods section. We believe this addition improves the statistical rigor and transparency of our manuscript (Please see lines 207-211, 221 and Table 1 in the revised text).
Comment: It will be possible the quantify and test the differences in the fluorescence test…
Response: We appreciate this suggestion. Our goal was to qualitatively confirm the juvenile testis-targeted gene transfer system. A robust quantification was beyond the scope of this protocol development study. Based on your suggestion, this point is described as limitation in the Discussion section of the revised text (Please see lines 308-313).
Comment: I suggest being more precise in the conclusions. Also, …discuss (in the discussion section) what is new in this methodology regarding other published experiments
Response: This sounds reasonable. We revised the Conclusions for greater precision. We also added a new paragraph to the Discussion section, in which our neonatal intra-testicular injection method is directly compared with the previously published methods, highlighting the novelty of applying this technique to a highly vulnerable age group (d 3-5) with a high survival rate (Please see lines 269-283 in the revised text).
Comment (Line 78-79): I suggest reporting the advantages and limitations of this equipment methodology… This information will support the justification for the objectives of your study.
Response: Thank you for your excellent suggestions. To strengthen the rationale of our study, we moved the phrase showing limitations of a previously reported device to the Introduction section (Please see lines 78-87 in the revised text).
Comment (Line 217): 2? It is not reported in Table 1.
Response: We thank the reviewer for pointing out this ambiguity. The super script “2” in Table 1 corresponds to a foot note explaining the cause of death. We revised the footnote for clarity (Please see lines 226 in the revised text).
Comment (Line 210-214): This two-day assessment was made in all 42 pups?
Response: Thank you for this important request for clarification. Yes, the two-day assessment was indeed performed on all 42 surviving pups. We apologize that the original text has been ambiguous on this point. We revised the sentence in the Materials and Methods section to explicitly state that all surviving animals were used for the fluorescence analysis, ensuring clarity for the reader (Please see lines 198-199 in the revised text).
Comment (line 217): Can you quantify instead of using almost?
Response: We agree that “almost” is imprecise. We revised this sentence to be more descriptive without implying quantification (Please see lines 235 in the revised text).
Comment (Line274-276): What was the success of these previous experiments
Response: This is a helpful point. We added the success rate from the cited paper (Ju et al.) in the Discussion section to provide better contest (Please see lines 269-275 in the revised text).
Round 2
Reviewer 2 Report
Comments and Suggestions for Authors
My concerns were addressed.
A minor issue: in Line 140-141, "Care must be taken regarding the volume of the anesthetic applied to the cotton." As a protocol paper, more details should be provided to improve reproducibility.
Author Response
We sincerely thank the reviewer for this important and helpful comment. We completely agree that the original phrase was too vague and that providing a specific, observable endpoint is crucial for ensuring the reproducibility of our protocol.
To address this, we have revised the manuscript to specify that the anesthetic depth was adjusted based on a standard and reliable indicator: the pup's response to a firm toe pinch (please see Lines 140 - 144 in the revised text).
We believe this provides a much clearer and more practical guide for researchers attempting to replicate our method.
Reviewer 3 Report
Comments and Suggestions for Authors
Dear authors,
Thank you for submitting this revised version to the journal. All comments and suggestions from this reviewer have been considered. The context/introduction and discussion have been refined, the replicability of the experiment has been ensured, and the conclusions are objective regarding the value of this methodology. Thank you.
In my opinion, no further revision is required. The current version can be published.
Author Response
We are very grateful to the reviewer for their positive and encouraging feedback on our revised manuscript. We are delighted that they found our revisions to be comprehensive and that the refinements made to the introduction and discussion, as well as the improved replicability of the methodology, have fully addressed their initial suggestions.
The reviewer's constructive guidance was instrumental in significantly strengthening our paper, and we sincerely appreciate their support for its publication.